# Dispersal Kernel Type Highly Influences Projected Relationships for Plant Disease Epidemic Severity When Outbreak and At-Risk Populations Differ in Susceptibility

**DOI:** 10.3390/life12111727

**Published:** 2022-10-28

**Authors:** Paul M. Severns

**Affiliations:** Department of Plant Pathology, University of Georgia, Athens, GA 30602, USA; paul.severns@uga.edu

**Keywords:** disease gradient, disease outbreak, *Puccinia*, wheat stripe rust, plant epidemic, dispersal ecology

## Abstract

In silico study of biologically invading organisms provide a means to evaluate the complex and potentially cryptic factors that can influence invasion success in scenarios where empirical studies would be difficult, if not impossible, to conduct. I used a disease event simulation program to evaluate whether the two most frequently used types of plant pathogen dispersal kernels for epidemiological projections would provide complementary or divergent projections of epidemic severity when the hosts in a disease outbreak differed from the hosts in the at-risk population in the degree of susceptibility. Exponential dispersal kernel simulations of wheat stripe rust (*Puccinia*
*striiformis* var *trittici*) predicted a relatively strong and dominant influence of the at-risk population on the end epidemic severity regardless of outbreak disease levels. Simulations using a modified power law dispersal kernel gave projections that varied depending on the amount of disease in the outbreak and some interactions were counter-intuitive and opposite of the exponential dispersal kernel projections. Although relatively straightforward, the disease spread simulations in the present study strongly suggest that a more biologically accurate dispersal kernel generates complexity that would not be revealed by an exponential dispersal gradient and that selecting a less accurate dispersal kernel may obscure important interactions during biological invasions.

## 1. Introduction

Complex systems are difficult to study empirically, but its components can be understood or at least statistically described in a way that the information can be used to create models to project responses under scenarios that may be impossible to create experimentally [1,2,3]. For invasion biology, models are an important tool for projecting the spatio-temporal patterns of a biological invasion, and they can also facilitate investigations into difficult to study factors and how they may suppress or encourage organism invasion. The insights gained from carefully constructed models containing well-established ties to biologically realistic mechanisms can be crucial for implementing mitigation strategies to control the invading organism [4,5]. While in silico studies are an obvious departure from on-the-ground empirical study and require simplifying assumptions, they are an important method to understand and project the impacts and spatio-temporal patterns associated with biological invasions. A wait and see strategy for the empirical study of a biological invasion is simply not pro-active enough given the world-wide loss of biodiversity, human life, and ecosystem changes that are now the text-book outcomes of uncontrolled biological invasions. 

To create a potentially useful statistical model of organism invasion, the stages of the invading organism in terms of colonization, reproduction, and dispersal need to be integrated and preferably run in a spatially explicit, virtual landscape [6]. Demographic rates (e.g., vital rates) and colonization probability can be measured through observation and/or estimated through direct empirical study, manipulative experimentation, and/or in combination with in silico methods that use sensitivity analyses and pattern-oriented modeling [2,6]. One of the more difficult, but critically important, aspects of biological invasion models to measure and/or estimate is the dispersal kernel. The dispersal kernel is a mathematical function that is used to statistically describe how an organism disperses through a landscape over time. Dispersal kernels are notoriously difficult to measure and to accurately parameterize for organisms that are prone to rare, long-distance dispersal events. The challenge to represent the rarer successful long-distance dispersal events are that the successful events are sparse and embedded within a large expanse of absences and this is type of data is information poor compared to the area near an invasion source which contains a relatively large number of successful dispersal events over shorter distances. Because these long-distance events are rare, they can be easily underestimated by a dispersal kernel but be biologically meaningful for the patterns of invasion spread [7].

Disease epidemics are considered a form of biological invasion [8] and they share similar factors that influence epidemic severity as well as control philosophies [9,10,11,12,13]. For plant pathogens, there are two primary types of dispersal kernels that have been used to project the spread of aerially vectored plant diseases (primarily fungal diseases). One family of dispersal kernels are those with functions that are exponentially bound (e.g., exponential, double exponential) and the other family consists of those functions that are not bound by an exponential (e.g., modified power law, modified Pareto distribution) [7,14]. Exponential functions have longer distributional tails (more kurtotic) than a normal distribution and describe the decrease of inoculum/disease dispersed from a source over a greater expanse with rapidly decreasing disease levels as the distance from the source increases. Exponential family dispersal kernels eventually terminate when either the fitted function to empirically collected data crosses the x-axis or the probability of occurrence reaches zero (for probability density functions). The non-exponentially bound dispersal kernels are comparatively more leptokurtic (fatter, thicker, or heavier tails) than the exponentially bounded functions, with the distribution’s tails extending for a much greater distance at very low predicted probabilities. In comparison, these more kurtotic functions expand the small probability of long-distance dispersal events over a much greater distance than exponential kernels. 

It is a mathematically demonstrated outcome that if the amount of host is approximately continuous and homogenously distributed, and in a sufficiently sized area, that exponentially bound functions will produce disease invasion fronts that move through the host population with a constant rate following a short period of acceleration [14,15]. These exponentially bound dispersal kernel functions simplify to a diffusion rate, a constant rate of disease spread over space, and this property facilitates straightforward predictive diffusion-based epidemiological projections. However, there is also empirical evidence that wind vectored plant diseases are inadequately described by an exponential function (the function’s tails are significantly truncated compared to the actual observed dispersal gradient) and that non-exponentially bound functions (dispersal kernels with much longer distribution tails) are biologically more appropriate [7,16,17,18]. In contrast to the exponential family of dispersal kernels, the long-tailed, non-exponentially bound dispersal kernels produce disease invasions with fronts that appear to always increase in velocity over space until host and/or space become limiting, and therefore cannot be represented by a rate constant, even as a simplification [14,16,18]. Provided the same raw data which were modeled under the same environmental (and host) conditions, these two dispersal kernel types not only generate different rates of organism spread but they also predict markedly different patterns of disease abundance with respect to its source [14,15,16,17].

The issue of disease susceptibility, especially as it pertains to understanding and projecting the spread of disease, is an important topic given that vaccinations are expected to generate specific outcomes in the at-risk population and disease resistance bred into plants should suppress disease. However, this issue is not straightforward to study empirically, as between field borders can differ in cultivar composition, fields may be intercropped, cultivar mixtures can be planted, and even alternating rows of different cultivars and fungicide treatments (a cost saving technique that lowers fungicide application rates) are not uncommon grower practices. For such scenarios, it reasonable to ask whether there is a suppressive or facilitative influence (and whether this impact may be predictable) on subsequent epidemic severity when disease disperses from the outbreak into an at-risk population where host resistance is either greater or lower than that of the outbreak host population. For the purposes of this manuscript, I consider the outbreak to be the area (and its host plants) that the initial disease generation occupies and the at-risk population to be all hosts outside of the outbreak. It is possible that the answer to this question could be purely demographic in nature - simply that the reduction or increase in relative reproductive rates is the primary determinant of later disease severity in an at-risk population. However, the shape and degree of dispersal kernel kurtosis can generate a strong impact on the subsequent patterns of spread and the spatial patterns of disease intensification from an outbreak as it spreads into the at-risk population [14,16,19,20,21].

I used a series of in silico experiments to understand whether the dispersal kernel type, exponentially bound or non-exponentially bound, substantively influences the patterns of disease projections when disease transitions between outbreak and at-risk host populations that differ in disease susceptibility. I focused on wheat stripe rust, an economically important, world-wide, disease of wheat caused by the fungus *Puccinia striiformis* var *tritici* (hereafter *Pst*), a well-studied and relatively well characterized plant pathosystem from an epidemiological perspective. In particular, I was interested in evaluating whether one or both dispersal kernel types (exponentially bound or non-exponentially bound) could yield relationships that are consistently predictable over a range of disease outbreak levels and whether those projections are similar enough to suggest that a simplified approximation could be made about the potential interactions. For example, it is possible that the overall difference in susceptibility between the outbreak and at-risk populations proportionally increases or decreases the amount of disease in the at-risk population according to a predictable linear relationship. 

## 2. Materials and Methods

### 2.1. Wheat Stripe Rust

Wheat stripe rust (WSR) is caused by the fungus *Puccinia striiformis* var *tritici* (*Pst*) and it is an obligate parasite of its host plant (obligate plant pathogens require relatively healthy and vigorous hosts for disease to occur). WSR can be encountered wherever wheat is grown [22,23,24,25] and its alternative host plants appear to be *Berberis* spp. [26,27]. However, it is unlikely that *Berberis* spp. are necessary for WSR epidemics as *Pst* spores can overwinter in the soil and thatch when conditions are mostly above freezing [23,28]. *Pst* produces spores (~10 to 20 microns which appear to be somewhat environmentally resilient to temperature and some UV light exposure [29]) that are borne on uredinia in small aggregates referred to as pustules. Groups of pustules form lesions, which are presented linearly on the upper and lower leaf surfaces, and elongate over time parallel to wheat leaf veins, yielding the “striped” appearance of WSR. Spores are produced in large amounts, several hundred or more uridineospores/day per square millimeter of lesion [25], and R_0_ (the basic reproductive number, the mean number of daughter infections arising from a single infection) can be very high (ranging from 35 to ~800) depending on host availability and pre-existing disease levels [30,31]. Disease occurs as long as the wheat plant can physiologically support either new infections or the expansion of existing lesions. As WSR outbreaks intensify, the disease grows at an exponential rate [10], but successful dispersal events can occur over large distances even from relatively small outbreaks [32] and rarer long-distance events are known at continental scales [28,33]. 

Although wheat stripe rust can be theoretically well-managed through the appropriate timing of fungicide applications [34], WSR epidemics can cause massive damage on susceptible wheat cultivars [23,24,28,35]. Unfortunately, there is also recent evidence that some *Pst* lineages have evolved fungicide resistant mutations [36], which has caused considerable problems for the management other wheat fungal diseases on wheat such as Septoria leaf blotch (*Zymoseptoria trittici*) [37,38,39,40], eyespot (*Oculimacula* spp.) [41], and wheat blast (*Magnaporthe oryzae*) [42]. With the increasing incidence of fungicide resistant wheat plant diseases across the world, including *Pst*, control will probably be accomplished through the breeding of durable disease resistance [43]. This means that understanding how disease susceptibility may alter epidemic behavior is an important aspect to understand going forward.

### 2.2. Wheat Stripe Rust Disease Spread Simulations

I used an updated and highly modified version of the plant disease simulation program EPIMUL [44] to run the in silico projected disease spread experiments. EPIMUL is a spatially explicit, compartmental disease event simulator, which is parameterized to represent real space in a wheat field. Each compartment was filled with virtual host plants that were similar in density to production fields and previously published empirical studies of WSR spread. Plants within the compartment were assigned properties (e.g., density, disease carrying capacity, latent and infectious periods, disease reproduction rates, infection probability, outbreak or at-risk population) and effective disease spores were distributed across this landscape according to a specified dispersal kernel. The epidemiological variables used in the present model originated from published intensive field studies performed in western and central Oregon, USA (see below for simulation and parameter details). For this study, I used deterministic simulations as I was interested in the mean differences between scenarios rather than focusing on the variation within a single scenario and how that variation overlaps with a slightly different set of parameter values. In EPIMUL, stochasticity is built into the dispersal gradient as a Poisson resampling of the original dispersal gradient [10]. In previous simulations, the mean disease levels over space and time in each compartment from 100 stochastic simulations was nearly equivalent to one deterministic run in EPIMUL [10], so while there was information in variability to be gained from stochastic simulations this approach was not necessary given the goals of the present study.

Compartment parameters for the simulations were consistent with previous WSR simulations [17,45] and updated with more accurate parameter values when supported by newer published data. The simulation field size was 800 × 800 compartments, with each compartment having dimensions of 1.52 m × 1.52 m (the width of a wheat planter) and each compartment had a carrying capacity of 200,000 infection sites, which is the average number of sites estimated from a standard wheat planting density over the life of the average wheat plant [17]. I used a latent and infectious period of 12 days, which is common for WSR outbreaks in the late spring and early summer when conditions are optimal for the disease in central Oregon. R_0_, the basic reproduction number [46], which is the mean number of daughter infections arising from a single mother infection, was set at 70 for the completely susceptible genotype and reduced proportionally with a decrease in susceptibility (an increase in disease resistance). This method is described below in a separate paragraph. The fully susceptible host R_0_ = 70 is consistent with previous experiments featuring fully susceptible and partially susceptible wheat genotypes [45,47] and studies of WSR development [30] over a range of environmental conditions that were comparable to central Oregon. 

I used two different dispersal kernels to simulate WSR disease spread in the exact same virtual field arrangement to understand the influence of each dispersal kernel type on epidemic projections. The first gradient was the modified power law dispersal kernel reported by Farber et al. [32]. This is the most accurately and precisely described dispersal gradient for WSR available in the published literature. For the modified power law, the dispersal kernel was described by the formula y = a (x + c)^−b^ where “a” was a value that adjusts the amount of disease produced at the source; b modified the steepness of the dispersal kernel; and the c value allowed for the power law dispersal kernel to have a non-zero value when x = 0 and also modified the kernel shape. For the modified power law simulations, the values of each variable were: a = 425, b = 2.28, c = 0.23. The exponential function was calculated from the original data used by Farber et al. [32] (which was originally and appropriately best-fit to the power law kernel above) and an exponential model was forced on the Farber et al. [32] raw data with the method traditionally used by plant pathologists to fit disease gradients to an exponential kernel [48]. The exponential dispersal kernel was described by the following formula: y = a exp (−bx); for this study a = 19.2, b = 0.1903. WSR infections were dispersed equally (radially) from the source using the downwind dispersal gradient reported by Farber et al. [32]. 

I also evaluated the potential influence of the amount of disease in the outbreak on the projections. Disease levels in the outbreak can have a strong and dominant impact on the severity of the subsequent WSR epidemic in the at-risk population, in field experiments [45,47] and in simulations [10]. It is possible there were dispersal kernel × outbreak disease level interactions that influence epidemic severity when host populations differ in disease susceptibility. The outbreak levels of disease in my simulations were set at 0.05%, 1.0%, and 5.0% of the total sites available (disease carrying capacity), and these values span the range of biologically reasonable outbreak levels (0.05% and 1.0%) and exceptionally high outbreak levels (5%).

I set up two virtual landscapes that were used with both dispersal kernels and each disease outbreak level to project the interactions of epidemic severity given the differences in host disease susceptibility in a standardized landscape. Both fields contained an outbreak (focus) that was one compartment (1.52 m × 1.52 m) in the center of an 800 × 800 compartment landscape. All compartments other than the outbreak represent the at-risk host population. In one scenario, the outbreak compartment was always 100% susceptible but the at-risk population host susceptibility varied in increments of 10% (from 100% to 10%). A susceptibility of 0 would not generate disease in the model as these hosts are completely resistant and useless in the present study. In the second scenario, the at-risk population was always 100% susceptible but the outbreak varied in susceptibility by increments of 10% (Figure 1). To compare the relative effect of the transition from populations of host that differed in susceptibilities, an internal control, I simulated disease spread in monocultures for the same increments of susceptibility (e.g., 10% focus to 10% at-risk, 50% focus to 50% at-risk, 100% focus to 100% at-risk). 

To model the differences in host susceptibility within the two landscape scenarios, I proportionally decreased R_0_ in 10% increments and assigned the desired levels of susceptibility to the outbreak and at-risk compartments (e.g., 100% susceptible hosts had an R_0_ = 70, 10% susceptible hosts have an R_0_ = 7). I held the infection probability the same for the dispersed effective spores which, in combination with a decreased R_0_, reduced their capacity for disease production if infected. Although, this approach is overly simplistic, as biologically resistance can arise from different mechanisms (e.g., reduced infection probability, reduced virulence, smaller lesions, lower sporulation rates), proportionally reducing the R_0_ is a straightforward method to represent hypothetical quantitative resistance from any mechanism and evaluate the resultant patterns of epidemic progression. To index the relative amount of disease that accumulated in the at-risk population from the outbreak after five disease generations (60 days), I calculated the area under the disease gradient (AUDG) for a 1 × 301 compartment area extending from the outbreak in a straight line (Figure 1). I subtracted the amount of disease in the outbreak compartment to arrive at an end epidemic AUDG value for the at-risk population. Calculating the amount of disease along a transect in the simulations mimics empirical studies of plant disease spread that sample disease at points along a straight line from the source [16,17,45,47,48]. I plotted the AUDG values for all different combinations of outbreak disease levels, monocultures, and disease accumulated in the at-risk populations and grouped the simulations by the two different landscape scenarios for comparison. 

## 3. Results

There were general patterns of disease increase that were consistent regardless of the dispersal model. Overall, the AUDG values (an index of relative epidemic severity) was predictably greater when the amount of disease in the outbreak was greater (Figure 2). Additionally, when the outbreak and at-risk populations were both at 100% susceptibility, the projected amount of disease was the greatest observed, and when either the outbreak or at-risk population was comprised of host plants with 10% susceptibility, epidemic severity was the lowest observed (Figure 2). However, projections from the two dispersal kernel types yielded differently shaped responses in the amount of disease accumulated after 5 generations.

The exponential dispersal kernel projected relatively consistent epidemic responses when disease developed from an outbreak and intensified over time in at-risk population which differed from the outbreak in the degree of disease susceptibility (Figure 2D–F). When the at-risk population was 100% susceptible, the at-risk population susceptibility exerted a dominant influence on the amount of disease that accumulated over time in the at-risk population, regardless of host susceptibility in the outbreak (Figure 2D–F orange lines). When the outbreak was 100% susceptible, the at-risk population degree of susceptibility also strongly influenced the amount of disease that accumulated in the at-risk population (Figure 2D–F blue lines). For both landscape scenarios, the projected relationships were approximately linear at the lower outbreak disease levels (0.05% and 1%), suggesting that host susceptibility of the at-risk population drives epidemic severity in a potentially straightforward and predictable manner. Only at the greatest outbreak disease level (5%), did the projected relationships become more curvilinear (Figure 2F), but the influence of the at-risk host population susceptibility on the end epidemic severity was consistent with lower outbreak disease levels. 

There was no consistent pattern of end epidemic severity when disease was dispersed with the modified power law (Figure 2A–C). For each outbreak disease level, the influence of either the outbreak or the at-risk population’s level of susceptibility appeared to generate different projections of end epidemic severity (Figure 2A–C). In all circumstances, including the monocultures, the projected epidemic severity relationships did not appear to behave in any obvious generalizable manner and different outbreak disease levels projected different relationships. For example, at the lowest outbreak disease level, 0.05%, the projected epidemic severities were strongly curvilinear while at the greatest outbreak disease level (5% disease) the projected relationships were more linear than curvilinear compared with lower outbreak disease levels. Unlike the exponential dispersal kernel simulations, the outbreak could either strongly influence the epidemic outcome (Figure 2A), be roughly equivalent in its influence to that of the at-risk population (Figure 2B), or be slightly suppressed by the level of susceptibility of the at-risk population (Figure 2C).

## 4. Discussion

WSR disease spread simulations, which were calibrated against well-characterized demographic, epidemiological and dispersal parameters, yielded conflicting projections of how disease susceptibility may alter epidemic severity when the outbreak and at-risk host populations differ in their degree of resistance. These differences emerged as an interaction between the dispersal kernel type and the amount of disease that founded the outbreak. Host arrangement, including the virtual field size, compartment number and its dimensions, the location of the outbreak within the virtual field, the area from which disease was estimated, the latent and infectious periods, host density, generation time, infection probability and R_0_ (for 100% susceptible genotypes) were standardized throughout the simulations. Disease was also dispersed in a radially symmetric manner from the focus (there was no asymmetric anisotropy; e.g., upwind, downwind, changing wind directions and magnitude) to keep the scenarios as straightforward as possible for comparison. Only the dispersal kernel, the amount of disease in the focus at the outbreak onset, and the susceptibility of host plants in specific compartments (through the proportional reduction of R_0_) were modified. Despite this degree of standardization and constant conditions that are obvious departures from a “real life” WSR outbreak, simulations suggested that rule of thumb guidelines for predicting where, when and how much disease may be generated may be possible for organisms with exponential dispersal kernels but unrealistic for organisms characterized by long-distance dispersal. The penalty for over-simplification (a truncated dispersal gradient) was a suite of facile but potentially seriously misleading epidemic projections. These projections were attractive for suggesting a potential predictable pattern of disease spread, whereas attempting to reflect a more biological realistic scenario (through a well-fit dispersal kernel) gave a less intuitive assemblage of epidemic projections. There appeared to be an important tradeoff threshold between convenient interpretation and attempting to reasonably represent the biological reality of long-distance dispersal due specifically to the dispersal kernel.

For the sake of disease management and projecting the impacts of having a mosaic landscape of hosts that differ in disease resistance, it would be convenient if WSR was dispersed according to an exponential function. If WSR dispersal was realistically approximated by an exponential type kernel, understanding and projecting WSR impacts would be relatively tractable, as the constant rate of disease spread after an initial and short period of increasing velocity [15] could be approximated by a diffusion rate [49]. Diffusion rate projections are often applied in invasive species models [7,50,51] as they are in plant epidemiological models [48,49,52]. Furthermore, the exponential dispersal kernel simulations consistently projected a dominant influence of the at-risk population disease resistance properties (susceptibility) on the end epidemic severity (Figure 2). Conceptually, disease resistance properties and host density within the at-risk population are the fundamental underpinnings and assumptions of effective modern disease management tactics, such as quarantine zones, vaccinations, and ring culls [1,53,54,55]. Although, the prioritization of the at-risk population to control disease outbreaks is intuitive, as on-the-ground approaches often prioritize protecting and modifying the at-risk population to contain and dampen the impacts of any disease outbreak, it may be only a partial solution [55].

In contrast to the relatively consistent and potentially straightforward projections of the exponential dispersal kernel simulations, the modified power law dispersal kernel projections were markedly variable and not intuitively predictable over the range of conditions evaluated. Modified power law kernel simulations suggested that the outbreak may generate a strong and dominant influence on the resulting epidemic severity when compared to the at-risk population, especially at low outbreak disease levels. These disease projection results are counter to most disease mitigation approaches which are directed towards treating and prioritizing the at-risk population (e.g., quarantines, vaccinations, ring culling). There is theoretical [56], empirical [10,45,47] and in silico support [45] for a dominant influence of the outbreak on the epidemic severity in the at-risk population, but the mechanisms governing this phenomenon are not yet well-understood. However, at higher outbreak disease levels, >1% of the total possible infections at the outbreak onset, the power law dispersal kernel projections suggested that the outbreak and at-risk population susceptibility properties may exert a roughly equal contribution to the end epidemic severity and at higher outbreak disease levels the at-risk population was projected to have a greater influence than the outbreak (Figure 2A vs. Figure 2B,C). These results suggest that the contributions of the outbreak and the at-risk population may be highly context dependent and challenging to predict if non-exponentially bound, heavy-tailed dispersal kernels are used to more realistically account for the potential of long-distance dispersal. This suggests that with increasing biological accuracy in the dispersal kernel, epidemic projections will likely become both more complex, context dependent, and unfortunately maybe necessarily nuanced.

The landscape scenarios I used in this study were straightforward (Figure 1) and relatively simple compared with the host spatial complexity of cultivar mixtures, intercropping, and a patchwork landscape mosaic of variably sized agricultural fields featuring different wheat cultivars interspersed with fields/habitats without WSR host plants. Regardless, when applying a modified power law dispersal kernel, which empirical data strongly supports over an exponential dispersal kernel [17,18,32], it is clear that the most biologically accurate of the two dispersal kernels yields unintuitive WSR disease projections even though there was no anisotropic disease dispersal (a well-known feature of wind dispersed pathogens [49,57]), homogenous host distribution, and host plants with invariant physiological states (both of which are not biologically true). As attractive, convenient, and readily interpretable as the disease projections from the exponential model appear to be, such biologically inaccurate models have the distinct potential to lead epidemiological understanding and the resulting control management practices down a deceptive path. 

## Figures and Tables

**Figure 1 life-12-01727-f001:**
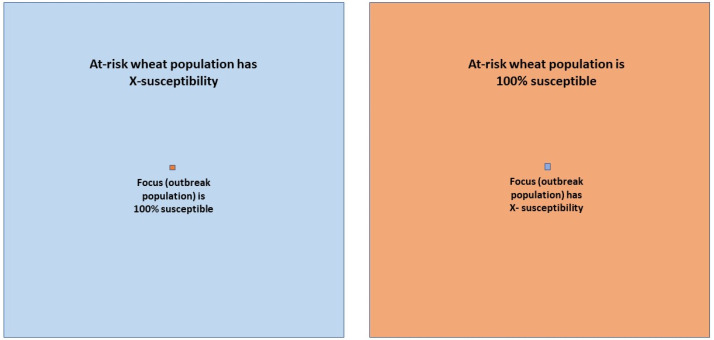
Schematic representation of two different landscape simulation scenarios where the outbreak (focus) and the at-risk population differed in the degree of quantitative resistance (susceptibility) by increments of 10% through the proportional reduction of R_0_ (see methods below). The left field depicts the scenario where the outbreak (focus) is comprised of a 100% WSR susceptible genotype and the at-risk population (the remainder of the field) decreases in the degree of susceptibility by increments of 10%. The right field depicts the opposite scenario where the focus is comprised of host plants that are variably susceptible and the at-risk population is 100% susceptible. Note that the focus and the at-risk field is not to the scale of the simulations. In the simulations the focus is considerably smaller relative to the at-risk field size.

**Figure 2 life-12-01727-f002:**
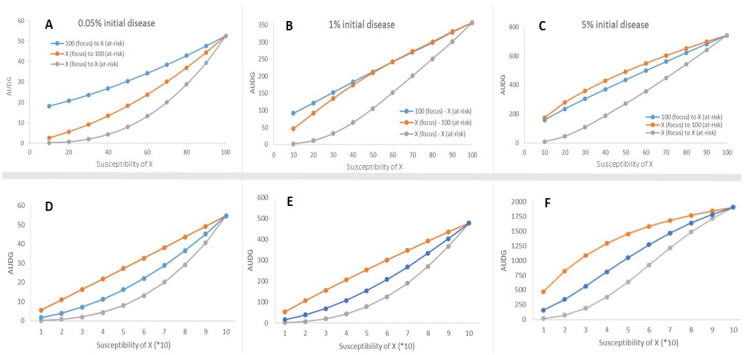
(**A**) (0.05% outbreak disease levels), (**B**) (1% outbreak disease levels), (**C**) (5% outbreak disease levels) are the trends projected from AUDG (area under the disease gradient) values generated from the two field scenario simulations where the outbreak and at-risk populations varied in their relative degree of susceptibility (quantitative resistance to WSR) using the modified power law dispersal kernel. (**D**) (0.05% outbreak disease levels), (**E**) (1% outbreak disease levels), (**F**) (5% outbreak disease levels) are the same projections and disease summary statistics that were generated by the simulations using the exponential dispersal kernel (note that the susceptibility is the same, but it is presented differently on the x-axis, with (*10) to draw attention that these figures were generated from the exponential dispersal kernel). Gray dots and trend lines represent simulation results from monocultures (e.g., 10% focus to 10% at-risk, to 100% focus to 100% at-risk), the blue dots and trend lines represent the scenario where the focus is 100% susceptible but the at-risk population has variable susceptibility, and the orange dots and trend line represents the scenario where the focus varies in susceptibility but the at-risk population is 100% susceptible.

## Data Availability

Data from the simulations are available upon request.

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
