# Peer review of "Dispersal Kernel Type Highly Influences Projected Relationships for Plant Disease Epidemic Severity When Outbreak and At-Risk Populations Differ in Susceptibility"

_life, 2022, doi:10.3390/life12111727_

Round 1
Reviewer 1 Report
Review of Severns, PM. Dispersal kernel type highly influences projected relationships for plant disease epidemic severity when outbreak and at-risk populations differ in susceptibility.
I enjoyed reading the Severns manuscript. The subject matter strikes me as important and interesting, and the study itself is well-designed and elegant in its design and execution. Overall, I think this work constitutes a valuable contribution to plant pathology and epidemiology, and the manuscript seems well matched to the journal Life.
Below I discuss a few ways in which the clarity of the overall message might be improved (major concerns), followed by a number of examples where minor edits would add clarity to individual passages (minor concerns). My numbering scheme is not intended to assign any degree of importance to the individual items. It is my expectation that these recommended changes could be completed with minor effort, simply by refining the manuscript text.
More Major Concerns
1. Throughout the manuscript, the author contrasts “outbreak hosts” with “at-risk hosts”. This is probably standard terminology, but I found it a bit confusing. Could such a functional distinction be made in the real world (where disease dynamics would also be included)? That is, in nature, wouldn’t the “outbreak hosts” simply be infected individuals, and the “at-risk hosts” be susceptible individuals? If so, those in the latter category would be continuously transitioning into the former. Because I tended to visualize the system this way, I found it difficult to imagine how the two categories of hosts could be intrinsically different. I realize that breeding or genetic engineering can produce wheat plants having different degrees of susceptibility to WSR. But wouldn’t entire wheat fields (or properties) tend to be planted with genetically similar plants? Perhaps a little clarification here, and maybe the introduction of a real world analog (e.g. a few lines of additional text) could help readers like myself better sort this out.
2. My understanding is that the study is designed to ask whether variations in disease susceptibility (the independent variable) will produce characteristic differences in the spatial distribution of infected and uninfected wheat plants (the response variable), when these outcomes are derived using two contrasting dispersal kernels (the experimental treatments). I found myself asking whether use of the two dispersal kernels would produce differences in the spatial distribution of infected and uninfected plants, even if disease susceptibility was not varied. Readers might benefit from understanding how the system responds to variation in only this single factor, prior to their grappling with variation in both dispersal kernel and host susceptibility simultaneously. Put another way, it seems like the simpler question (how does pattern vary with dispersal kernel) might precede the more complex question (how does pattern vary with dispersal kernel x host susceptibility).
3. The study’s results are essentially all derived from Figure 2. I must confess that I had a hard time fully reproducing the logic the author used to extract the paper’s key insights from this figure. To be clear, this reflects shortcomings in my expertise, not in the study. Nevertheless, I have a suggestion for how the manuscript’s results might be made easier for readers like myself to unpack.
What I think would have helped me the most would have been the addition of a thought experiment (or equivalent) that generated a figure that preceded Figure 1. This new illustration would be used to help the reader understand what the results should look like if the system responded in any of the various anticipated ways. I’m envisioning a cartoon that utilizes some artistic license (exaggeration) to help make the points clear. The new accompanying text would help the reader understand how to connect the results (in the format that you’ll present them later) to the processes that produce those results (what we're trying to gain insight into). I’m envisioning a discussion in which you introduce the significance of the blue line (hypothetically) moving below the orange line, and take the time to explain why all the curves should connect when X = 100, and so on. I could also have used some help understanding why panels C and F were different in some important way (or not), or even panels A and D, etc.
4. Two less substantial issues that confused me a bit:
(a) You state “WSR infections were dispersed equally (radially) from the source using a downwind dispersal gradient from the outbreak and I did not attempt to account for anisotropy as this adds confounding spatial interactions which are well beyond the scope of this manuscript.”
I had trouble understanding how propagules could be dispersed equally/radially while at the same time responding to a wind gradient. And wouldn’t a wind gradient introduce anisotropy? Some clarification here would help.
(b) You state “To index the relative amount of disease that accumulated from the outbreak after five disease generations (60 days), I calculated the area under the disease gradient (AUDG) for a 1 × 301 compartment area extending from the outbreak in a straight line (Figure 1) and subtracted the disease in the out-break compartment to arrive at an end epidemic AUDG value.”
It isn’t clear to the reader why the entire landscape wasn’t sampled instead of this 1 x 301 compartment subset. I’m sure there’s a good reason. But also, the way this is phrased, the reader may assume that the sampling area is displayed in Figure 1 (it isn’t).
More Minor Concerns
5. The parenthetical near the beginning of the abstract is a bit awkward, and could probably be deleted. This may provide unnecessary detail about the distributions, at least for this point in the paper.
6. In the beginning of the Introduction, you say “... can be used to create statistical models…” You then go on to describe mechanistic models. Do you want to say “statistical and mechanistic models”, or equivalent?
7. Perhaps change “the stages of the invading organism in terms of colonization, reproduction, and dispersal need to be integrated and preferably run in a spatially-explicit virtual landscape” to "the key stages exhibited by an invading organism, including colonization, reproduction, and dispersal, should be captured within a spatially-explicit simulation landscape".
8. You state “Dispersal kernels are notoriously difficult to measure and to accurately parameterize for organisms that are prone to rare, long-distance dispersal events as such events are important to represent from a biological perspective but difficult to capture statistically because such events are information sparse (as opposed to near an invasion source which contains a relatively large number of individuals dispersed frequently over shorter spaces)”. This could be shortened, possibly to something like “Dispersal kernels are notoriously difficult to obtain, especially for organisms that exhibit rare long-distance movement events”.
9. You say that “Exponential functions have longer distributional tails… than does a normal distribution”. This is the only time in the paper that a normal distribution is invoked, and I think it’s kind of distracting (the important contrast being made is between the shape of exponential and modified power law distributions).
10. You have a paragraph that begins with “The issue of disease susceptibility, especially as it pertains to understanding and projecting the spread of disease is an important topic”. This paragraph is a bit tough to get through, due to wording. But more importantly, I don’t think the reader will clearly understand that what’s going on here is your giving the background for how disease susceptibility can vary within a crop, which then is key to the methods employed in the experiment. So setting the stage for this information would be helpful.
11. Last sentence of the Introduction. Should “projects” really be “projections”?
12. Section 2.1 could be moved to the Introduction. I don’t have strong feelings about this, it’s just a suggestion to consider.
13. You say “Pst produces spores (~ 10 to 20 microns which appear to be somewhat environmentally resilient to tem-perature and some UV light exposure [29]) that are borne on uredinia in small aggregates referred to as pustules”. Perhaps this can be broken into two sentences?
14. You state “Unfortunately, there is recent evidence that some Pst lineages have evolved fungicide resistant mutations [36], and this type of resistance has caused considerable problems for the management other wheat fungal diseases on wheat such as Septoria leaf blotch (Zymoseptoria trittici) [37-40], eyespot (Oculimacula spp) [41], and wheat blast (Magnaporthe oryzae) [42]”. It wasn’t clear to me why the presence of fungicide-resistant Pst would complicate the management of other wheat pathogens. Do farmers quit applying fungicides, and then fall victim to a variety of plant pathogens?
15. In section 2.2, you state “...which is the mean number of daughter infections arising from a single mother infection…”. This information was already included in the first paragraph of section 2.1.
16. In section 2.2, where you describe the formulas for the power function and exponential function. You don’t actually state what X and Y refer to (it’s kind of obvious, but maybe you want to state it anyway?). There’s no mention of units (or lack thereof), “steepness” isn’t defined (again, it's kinda obvious but also a little jargony), and “finite” would be better replaced with “nonzero”.
17. There’s material in the first sentence of the caption for Figure 1 that describes material introduced for the first time below where that figure is placed. That’s a bit awkward, so you might just replace it with “(see Methods)” or equivalent.
18. The material in the caption of Figure 1 (which I mentioned above) is addressed in the first paragraph below Figure 1. There, you include the text “(e.g. 100% susceptible hosts had an R0 = 70, 10% susceptible hosts have an R0 = 7)”. I never did figure out why R0 was 70 when hosts were 100% susceptible (versus 66 or 88, etc.). I may have just missed something, but it didn’t appear obvious to me.
19. I had a lot of trouble with the second paragraph of the Results. It’s a bit rough.
20. I found the last sentence of the first paragraph of the Discussion to be particularly long. Perhaps it could be split into two sentences?
Author Response
I found Reviewer #1 comments to be very helpful in generating a better manuscript! Thank you for your efforts! I have responded to each comment in the space below following the original suggestion and included excerpts of the revised sections.
Review of Severns, PM. Dispersal kernel type highly influences projected relationships for plant disease epidemic severity when outbreak and at-risk populations differ in susceptibility.
I enjoyed reading the Severns manuscript. The subject matter strikes me as important and interesting, and the study itself is well-designed and elegant in its design and execution. Overall, I think this work constitutes a valuable contribution to plant pathology and epidemiology, and the manuscript seems well matched to the journal Life.
Below I discuss a few ways in which the clarity of the overall message might be improved (major concerns), followed by a number of examples where minor edits would add clarity to individual passages (minor concerns). My numbering scheme is not intended to assign any degree of importance to the individual items. It is my expectation that these recommended changes could be completed with minor effort, simply by refining the manuscript text.
More Major Concerns
- Throughout the manuscript, the author contrasts “outbreak hosts” with “at-risk hosts”. This is probably standard terminology, but I found it a bit confusing. Could such a functional distinction be made in the real world (where disease dynamics would also be included)? That is, in nature, wouldn’t the “outbreak hosts” simply be infected individuals, and the “at-risk hosts” be susceptible individuals? If so, those in the latter category would be continuously transitioning into the former. Because I tended to visualize the system this way, I found it difficult to imagine how the two categories of hosts could be intrinsically different. I realize that breeding or genetic engineering can produce wheat plants having different degrees of susceptibility to WSR. But wouldn’t entire wheat fields (or properties) tend to be planted with genetically similar plants? Perhaps a little clarification here, and maybe the introduction of a real world analog (e.g. a few lines of additional text) could help readers like myself better sort this out.
Response: I hopefully clarified this issue by defining the outbreak population as the area of hosts with disease in the first generation and the at-risk population as the entirety of the host population outside of the outbreak region. The definitions are based on time and space, specifically within the first disease generation. The definitions are stated in the Introduction, in the penultimate paragraph and read as follows. “For the purposes of this manuscript, I consider the outbreak to be the area (and its host plants) that the initial disease generation occupies and the at-risk population to be all hosts outside of the outbreak.” I also provided some instances where the in-field scenarios are not necessarily standard (all one cultivar) to attest to the potentially complicated nature of empirical study. The added sentence (also in the same paragraph) reads, “However, this issue is not straightforward to study empirically, as between field borders can differ in cultivar composition, fields may be intercropped, cultivar mixtures can be planted, and even alternating rows of different cultivars and fungicide treatments (a cost saving technique that lowers fungicide application rates) are not uncommon grower practices.”
- My understanding is that the study is designed to ask whether variations in disease susceptibility (the independent variable) will produce characteristic differences in the spatial distribution of infected and uninfected wheat plants (the response variable), when these outcomes are derived using two contrasting dispersal kernels (the experimental treatments). I found myself asking whether use of the two dispersal kernels would produce differences in the spatial distribution of infected and uninfected plants, even if disease susceptibility was not varied. Readers might benefit from understanding how the system responds to variation in only this single factor, prior to their grappling with variation in both dispersal kernel and host susceptibility simultaneously. Put another way, it seems like the simpler question (how does pattern vary with dispersal kernel) might precede the more complex question au(how does pattern vary with dispersal kernel x host susceptibility).
Response: I attempted to convey this generalized difference in terms of the type of invasion front that the disease produces if the pathogen/biological entity is distributed according to an exponentially bound function or a non-exponentially bound function with longer distribution tails. The papers cited demonstrate this and they show that the abundance and distribution of the invading organism is also different. I have modified the paragraph initially describing the travelling and dispersive waves of organism spread to indicate that projected and realized abundance of an organism over space and time also differs between the two dispersal kernels. I added the following sentence to the end of the 4th paragraph of the Introduction, “Provided the same raw data which were modeled under the same environmental (and host) conditions, these two dispersal kernel types not only generate different rates of organism spread but they also predict markedly different patterns of disease abundance with respect to its source [14-17].”
- The study’s results are essentially all derived from Figure 2. I must confess that I had a hard time fully reproducing the logic the author used to extract the paper’s key insights from this figure. To be clear, this reflects shortcomings in my expertise, not in the study. Nevertheless, I have a suggestion for how the manuscript’s results might be made easier for readers like myself to unpack.
What I think would have helped me the most would have been the addition of a thought experiment (or equivalent) that generated a figure that preceded Figure 1. This new illustration would be used to help the reader understand what the results should look like if the system responded in any of the various anticipated ways. I’m envisioning a cartoon that utilizes some artistic license (exaggeration) to help make the points clear. The new accompanying text would help the reader understand how to connect the results (in the format that you’ll present them later) to the processes that produce those results (what we're trying to gain insight into). I’m envisioning a discussion in which you introduce the significance of the blue line (hypothetically) moving below the orange line, and take the time to explain why all the curves should connect when X = 100, and so on. I could also have used some help understanding why panels C and F were different in some important way (or not), or even panels A and D, etc.
Response: I agree this would be ideal, but I did not anticipate the responses in the simulations, which is one of the reasons why I wanted to write this manuscript. To address the reviewer’s concerns, at least partially, I added the following sentence to the last paragraph of the Introduction, “For example, it is possible that the overall difference in susceptibility between the outbreak and at-risk populations proportionally increases or decreases the amount of disease in the at-risk population according to a predictable linear relationship.” This was unlikely to be true, but it was a possibility that may have occurred in an ideal simplistic world.
- Two less substantial issues that confused me a bit:
(a) You state “WSR infections were dispersed equally (radially) from the source using a downwind dispersal gradient from the outbreak and I did not attempt to account for anisotropy as this adds confounding spatial interactions which are well beyond the scope of this manuscript.”
I had trouble understanding how propagules could be dispersed equally/radially while at the same time responding to a wind gradient. And wouldn’t a wind gradient introduce anisotropy? Some clarification here would help.
Response: This was poor communication on my behalf. I simplified the text to read, “WSR infections were dispersed equally (radially) from the source using the downwind dispersal gradient reported by Farber et al. [32].”
(b) You state “To index the relative amount of disease that accumulated from the outbreak after five disease generations (60 days), I calculated the area under the disease gradient (AUDG) for a 1 × 301 compartment area extending from the outbreak in a straight line (Figure 1) and subtracted the disease in the out-break compartment to arrive at an end epidemic AUDG value.”
It isn’t clear to the reader why the entire landscape wasn’t sampled instead of this 1 x 301 compartment subset. I’m sure there’s a good reason. But also, the way this is phrased, the reader may assume that the sampling area is displayed in Figure 1 (it isn’t).
Response: Good point, thank you for bringing this to my attention. I added a sentence which explained that the disease measured along a transect from a source is how disease is monitored in field experiments. The sentence in the paragraph immediately preceding the results section now reads, “Calculating the amount of disease along a transect in the simulations mimics empirical studies of plant disease spread that sample disease at points along a straight line from the source [16,17,45,47,48].”
More Minor Concerns
- The parenthetical near the beginning of the abstract is a bit awkward, and could probably be deleted. This may provide unnecessary detail about the distributions, at least for this point in the paper.
Response: I have omitted the parenthetical statement from the abstract.
- In the beginning of the Introduction, you say “... can be used to create statistical models…” You then go on to describe mechanistic models. Do you want to say “statistical and mechanistic models”, or equivalent?
Response: I omitted “statistical” from the statement as it is true the term was redundant.
- Perhaps change “the stages of the invading organism in terms of colonization, reproduction, and dispersal need to be integrated and preferably run in a spatially-explicit virtual landscape” to "the key stages exhibited by an invading organism, including colonization, reproduction, and dispersal, should be captured within a spatially-explicit simulation landscape".
Response: I thank the reviewer for the suggestion, but I prefer to keep the original wording as there are other well-known plant disease models that run outside of a landscape event simulator framework yet attempt to integrate all of the same variables and parameter values.
- You state “Dispersal kernels are notoriously difficult to measure and to accurately parameterize for organisms that are prone to rare, long-distance dispersal events as such events are important to represent from a biological perspective but difficult to capture statistically because such events are information sparse (as opposed to near an invasion source which contains a relatively large number of individuals dispersed frequently over shorter spaces)”. This could be shortened, possibly to something like “Dispersal kernels are notoriously difficult to obtain, especially for organisms that exhibit rare long-distance movement events”.
Response: I rewrote this section and hopefully made the challenge in generating a more biologically accurate dispersal gradient as the lack of an information rich data set does not complement the biological impact of long-distance dispersal. The following passage now reads, “Dispersal kernels are notoriously difficult to measure and to accurately parameterize for organisms that are prone to rare, long-distance dispersal events. The challenge to represent the rarer successful long-distance dispersal events are that the successful events are sparse and embedded within a large expanse of absences and this is type of data is information poor compared to the area near an invasion source which contains a relatively large number of successful dispersal events over shorter distances. Because these long-distance events are rare, they can be easily underestimated by a dispersal kernel but be biologically meaningful for the patterns of invasion spread [7].”
- You say that “Exponential functions have longer distributional tails… than does a normal distribution”. This is the only time in the paper that a normal distribution is invoked, and I think it’s kind of distracting (the important contrast being made is between the shape of exponential and modified power law distributions).
Response: I agree, I thought it was distracting as well. But, normal distributions have been used to model plant disease spread by plant pathologists (even within the last 5 years), so I felt like that comparison needed to be made.
- You have a paragraph that begins with “The issue of disease susceptibility, especially as it pertains to understanding and projecting the spread of disease is an important topic”. This paragraph is a bit tough to get through, due to wording. But more importantly, I don’t think the reader will clearly understand that what’s going on here is your giving the background for how disease susceptibility can vary within a crop, which then is key to the methods employed in the experiment. So setting the stage for this information would be helpful.
Response: I think this concern was hopefully addressed by my response to concern #1 above.
- Last sentence of the Introduction. Should “projects” really be “projections”?
Response: Yes! Thank you very much!
- Section 2.1 could be moved to the Introduction. I don’t have strong feelings about this, it’s just a suggestion to consider.
- You say “Pst produces spores (~ 10 to 20 microns which appear to be somewhat environmentally resilient to tem-perature and some UV light exposure [29]) that are borne on uredinia in small aggregates referred to as pustules”. Perhaps this can be broken into two sentences?
Response: I performed a minor edit to this passage and hopefully improved it.
- You state “Unfortunately, there is recent evidence that some Pst lineages have evolved fungicide resistant mutations [36], and this type of resistance has caused considerable problems for the management other wheat fungal diseases on wheat such as Septoria leaf blotch (Zymoseptoria trittici) [37-40], eyespot (Oculimacula spp) [41], and wheat blast (Magnaporthe oryzae) [42]”. It wasn’t clear to me why the presence of fungicide-resistant Pst would complicate the management of other wheat pathogens. Do farmers quit applying fungicides, and then fall victim to a variety of plant pathogens?
Response: My rationale is that fungicides don’t protect crops once the pathogens have evolved resistance but that breeding resistant cultivars is probably the way forward to protect plants from diseases. I added a small piece of information to the following sentence, which I hope clarifies my intention.
It now reads, “With the increasing incidence of fungicide resistant wheat plant diseases across the world, including Pst, control will probably be accomplished through the breeding of durable disease resistance [43]. This means that understanding how disease susceptibility may alter epidemic behavior is an important aspect to understand going forward.”
- In section 2.2, you state “...which is the mean number of daughter infections arising from a single mother infection…”. This information was already included in the first paragraph of section 2.1.
Response: I repeated this definition intentionally, because there are often different definitions of Ro invoked by authors and sometimes the same authors use different definitions even in the same paper. I was attempting to be transparent at the detriment of repetition. I prefer to keep this repetition, albeit annoying.
- In section 2.2, where you describe the formulas for the power function and exponential function. You don’t actually state what X and Y refer to (it’s kind of obvious, but maybe you want to state it anyway?). There’s no mention of units (or lack thereof), “steepness” isn’t defined (again, it's kinda obvious but also a little jargony), and “finite” would be better replaced with “nonzero”.
Response: Yes, thank you!
- There’s material in the first sentence of the caption for Figure 1 that describes material introduced for the first time below where that figure is placed. That’s a bit awkward, so you might just replace it with “(see Methods)” or equivalent.
Response: Good suggestion. Thank you.
- The material in the caption of Figure 1 (which I mentioned above) is addressed in the first paragraph below Figure 1. There, you include the text “(e.g. 100% susceptible hosts had an R0 = 70, 10% susceptible hosts have an R0 = 7)”. I never did figure out why R0 was 70 when hosts were 100% susceptible (versus 66 or 88, etc.). I may have just missed something, but it didn’t appear obvious to me.
Response: This is the Ro value that has been calculated for wheat stripe rust in several different studies and the citations for these studies were provided in the methods section [references 45 and 47].
- I had a lot of trouble with the second paragraph of the Results. It’s a bit rough.
Response: I edited this paragraph substantially and hopefully it is now clearly written. It reads, “The exponential dispersal kernel projected relatively consistent epidemic responses when disease developed from an outbreak and intensified over time in at-risk population which differed from the outbreak in the degree of disease susceptibility (Figure 2 D-F). When the at-risk population was 100% susceptible, the at-risk population susceptibility exerted a dominant influence on the amount of disease that accumulated over time in the at-risk population, regardless of host susceptibility in the outbreak (Figure 2 D-F orange lines). When the outbreak was 100% susceptible, the at-risk population degree of susceptibility also strongly influenced the amount of disease that accumulated in the at-risk population (Figure 2 D-F blue lines). For both landscape scenarios, the projected relationships were approximately linear at the lower outbreak disease levels (0.05% and 1%), suggesting that host susceptibility of the at-risk population drives epidemic severity in a potentially straightforward and predictable manner. Only at the greatest outbreak disease level (5%), did the projected relationships become more curvilinear (Figure 2 F), but the influence of the at-risk host population susceptibility on the end epidemic severity was consistent with lower outbreak disease levels.”
- I found the last sentence of the first paragraph of the Discussion to be particularly long. Perhaps it could be split into two sentences?
Response: I rewrote this sentence and hopefully clarified several points which were not well articulated. The passage now reads, “The penalty for over-simplification (a truncated dispersal gradient) was a suite of facile but potentially seriously misleading epidemic projections. These projections were attractive for suggesting a potential predictable pattern of disease spread, whereas attempting to reflect a more biological realistic scenario (through a well-fit dispersal kernel) gave a less intuitive assemblage of epidemic projections. There appeared to be an important tradeoff threshold between convenient interpretation and attempting to reasonably represent the biological reality of long-distance dispersal due specifically to the dispersal kernel.”
Reviewer 2 Report
proof read the manuscript multiple times before final submission. consider the following articles
1. Automated Plant Leaf Disease Detection and Classification Using Fuzzy Based Function Network
2. Leaf Disease Segmentation and Classification of Jatropha Curcas L. and Pongamia Pinnata L. Biofuel Plants using Computer Vision based approaches
3. Web facilitated Anthracnose disease segmentation from the leaf of Mango tree using Radial basis function (RBF) neural network
4. Applications of Computer Vision in Plant Pathology: A Survey
Author Response
proof read the manuscript multiple times before final submission. consider the following articles
Response: Thank you, I have edited the manuscript and made multiple editorial and grammatical changes throughout the manuscript. I am not sure what the reviewer wanted me to understand and integrate from the listed studies below.
1. Automated Plant Leaf Disease Detection and Classification Using Fuzzy Based Function Network
2. Leaf Disease Segmentation and Classification of Jatropha Curcas L. and Pongamia Pinnata L. Biofuel Plants using Computer Vision based approaches
3. Web facilitated Anthracnose disease segmentation from the leaf of Mango tree using Radial basis function (RBF) neural network
4. Applications of Computer Vision in Plant Pathology: A Survey
Reviewer 3 Report
Title- Go through the title again as it sounds more like a concluding sentence.
Abstract - Check whether "I" is used to explain the work done in the research article or not , as "I" used in the abstract and introduction part sound like you are sharing your experience s rather than explaining your work done.
Introduction- Concentrate on improvement in terms of grammar and spellings. e.g. ' diseasing carrying capacity' should be written as ' disease carrying capacity'.
Materials and Methods- The procedure should be explained in past tense as you are explaining your work that you have already done.
Results and discussion- Concentrate on improvement in terms of grammar as there are some grammatical errors.
References- All the references mentioned in the text are present in reference section. Check the format of references as per the guideline mentioned by the journal.
Author Response
Reviewer #3
Title- Go through the title again as it sounds more like a concluding sentence.
Response: This title was intentional on my part. I wanted the title to be a statement which would capture the audiences of plant epidemiologists, landscape ecologists, and invasion biologists.
Abstract - Check whether "I" is used to explain the work done in the research article or not , as "I" used in the abstract and introduction part sound like you are sharing your experience s rather than explaining your work done.
Introduction- Concentrate on improvement in terms of grammar and spellings. e.g. ' diseasing carrying capacity' should be written as ' disease carrying capacity'.
Response: Thank you, I reread the section and found a few edits like the one suggested above.
Materials and Methods- The procedure should be explained in past tense as you are explaining your work that you have already done.
Response: Yes agreed, thank you for the suggestion! I have been careful in the revised version about the verb tenses used throughout the manuscript.
Results and discussion- Concentrate on improvement in terms of grammar as there are some grammatical errors.
Response: I have caught and correct several grammatical and stylistic errors in the revision that hopefully improves this manuscript.